# SELF-SUPERVISED PRETRAINING FOR DIFFERENTIALLY PRIVATE LEARNING

## ABSTRACT

We demonstrate self-supervised pretraining (SSP) is a scalable solution to deep learning with differential privacy (DP) regardless of the size of available public datasets in image classification. When facing the lack of public datasets, we show the features generated by SSP on only one single image enable a private classifier to obtain a much better utility than the non-learned handcrafted features under the same privacy budget. When a moderate or large size public dataset is available, the features produced by SSP greatly outperform the features trained with labels on various complex private datasets under the same private budget. We also compared multiple DP-enabled training frameworks to train a private classifier on the features generated by SSP.

## 1 INTRODUCTION

Machine learning (ML) has been applied ubiquitously in the analysis of sensitive data such as medical images (Tajbakhsh et al., 2016), financial records (Fischer & Krauss, 2018), or social media channels (Agrawal & Awekar, 2018). Many attacks (Shokri et al., 2017; Carlini et al., 2021) are developed to successfully extract meaningful training data out of standard ML models. According to recent governmental regulations, e.g., GDPR and CCPA, ML models have to protect sensitive training data. *Differential privacy* (DP) (Chaudhuri et al., 2011; Bu et al., 2020; Abadi et al., 2016) has emerged as an effective framework to train models resilient to private training data leakage.

Unfortunately, training models with strong DP guarantees significantly hurts the model utility (i.e., accuracy) (Papernot et al., 2018; Abadi et al., 2016). Although non-learned handcrafted features such as ScatterNet (Oyallon & Mallat, 2015; Oyallon et al., 2018) make a private linear model (Tramer & Boneh, 2021) achieve the state-of-the-art (SOTA) utility of $< 70\%$ under the privacy budget of ($\epsilon \le 3, \delta = 10^{-5}$) on a private CIFAR-10 dataset, it is difficult to learn better features in the DP domain, since the clipped and perturbed gradients during DP training provide only a noisy estimate of the update direction. In contrast, it is straightforward that pretrained features learned from large public labeled (Luo et al., 2021) datasets can greatly mitigate the utility gap between private and non-private models. However, sometimes there is no available public dataset for training a feature extractor due to legal causes or ethical issues (Flanders, 2009).

In this paper, we aim to demonstrate that *self-supervised pretraining* (SSP) *is a scalable solution to improving the utility of deep learning with DP regardless of the size of available public datasets* in image classification. Any updates on the learnable parameters of a differentially private model increase privacy overhead. It is easier to achieve both high utility and small privacy loss via the features generated by a well-trained feature extractor that can fully take advantage of SOTA network architectures and public datasets. Even when no large public dataset is available, we show a feature extractor built upon data-efficient HarmonicNet (Ulicny et al., 2019) and trained by self-supervised SimCLRv2 (Chen et al., 2020) on *only one single image* (YM. et al., 2020) can make a private linear classifier obtain much better utility than the non-learned handcrafted features (Tramer & Boneh, 2021) under the same privacy budget. With a larger public dataset, the features generated by SSP substantially outperform the features trained with labels on various complex private datasets, as shown in Table 1. To better explore the trade-off between utility and privacy, we compared SOTA DP-enabled training frameworks, i.e., DP stochastic gradient descent (DPSGD) (Abadi et al., 2016), DP direct feedback alignment (DPDFA) (Ohana et al., 2021; Lee & Kifer, 2020), DP stochastic gradient Langevin dynamics (DPSGLD) (Bu et al., 2021), and Private Aggregation of Teacher En-

Table 1: The utility and privacy comparison on a private CIFAR10 dataset. DP-SOTA-1: Tramer & Boneh (2021); and DP-SOTA-2: Luo et al. (2021).

| Public Dataset | Scheme | $\epsilon$-DP | Network | Training | Accuracy (%) |
|---|---|---|---|---|---|
| None | DP-SOTA-1 | 3 | Scat + CNN | DPSGD | 66.1[1] |
| 1 image | **Ours** | 3 | **HarmRN18** | DPSGD | **66.3**[2] |
| labeled CIFAR100 | DP-SOTA-2 | 1.5 | ResNet18 | DPSGD | 71[1] |
| **unlabeled** CIFAR100 | **Ours** | 1.5 | ResNet18 | DPSGD | **75.5**[2] |
| unlabeled ImageNet | DP-SOTA-1 | 2 | ResNet50 | DPSGD | 92.7 |
| | **Ours** | 2 | ResNet50 | **DPDFA** | **94.9** |

sembles (PATE) (Papernot et al., 2018), when training a private classifier on the features produced by SSP. *Private datasets having different learning distances from the public dataset favor different training frameworks under different privacy budgets.* Our contributions are summarized as:

- When facing the lack of public datasets, we adopt HarmonicNet as the backbone of SimCLRv2 to learn only one image. The features extracted by the HarmonicNet greatly outperform the non-learned handcrafted ScatterNet features on various private datasets by 0.6% on CIFAR10, 1.4% on CIFAR100, 6.7% on CropDiseases, 49.7% on EuroSAT, and 3.5% on ISIC2018, when $\epsilon = 2$.

- When there is a moderate or large size public dataset, the features produced by SSP improve the utility of these private complex datasets over the features trained with labels by $0.5\% \sim 8.6\%$ under the privacy budget of $\epsilon = 2$.

- We compared SOTA DP-enabled training frameworks, i.e., DPSGD, DPDFA, DPSGLD, and PATE, to train a private classifier on the features produced by SSP. Compared to DPSGD, DPSGLD obtains a better utility on private datasets when $\epsilon \leq 1$. DPDFA achieves a higher utility than DPSGD on private datasets with a smaller learning distance from the public dataset when $\epsilon > 0.5$.

## 2 BACKGROUND

### 2.1 DIFFERENTIALLY PRIVATE LEARNING

**Differential privacy**. A network model $M : \mathcal{D} \to \mathcal{R}$ is trained on two datasets $D, D' \in \mathcal{D}$, which differ only by a single data record. For any subset of outputs $R \in \mathcal{R}$, the model satisfies $(\epsilon, \delta)$-differential privacy (DP) (Abadi et al., 2016) if

$$\mathbf{Pr}[M(D) \in R] \leq e^\epsilon \cdot \mathbf{Pr}[M(D') \in R] + \delta$$

In another word, $\epsilon$ bounds the privacy loss on any single sample, and $\delta$ is the probability that this bound does not hold. Rényi DP (RDP) (Mironov, 2017) is a generalization of $(\epsilon, \delta)$-DP that uses Rényi divergence as a distance metric. More RDP details can be viewed in Appendix A.1.

**DP-enabled training**. DP guarantees can be enforced into a private network model by various DP-enabled training frameworks including DPSGD (Abadi et al., 2016), DPSGLD (Bu et al., 2021), DPDFA (Ohana et al., 2021; Lee & Kifer, 2020), and PATE (Papernot et al., 2018).

- **DPSGD**. DPSGD (Abadi et al., 2016) is the most widely used DP-enabled training framework that satisfies RDP. RDP during DPSGD on a model is enforced by two components: (1) per-sample gradients are clipped at a fixed L2 norm threshold $C$; and (2) Gaussian noise of magnitude $\sigma^2 C^2$ is added to the gradient updates for a noise scale parameter $\sigma$. The privacy cost $\epsilon$ during DPSGD can be measured by Moments Accountant (Abadi et al., 2016), which computes the upper bound of $\epsilon$ as a consequence of using different composition theories. The privacy loss rate depends on the hyper-parameters (Mohapatra et al., 2021) of DPSGD.

- **DPSGLD**. SGLD (Welling & Teh, 2011) is a gradient technique to train Bayesian networks. SGLD makes the weights of a model to converge to a posterior distribution rather than to a

---

[1]Group normalization. The results of batch normalization can be found in Appendix A.2.
[2]Batch normalization trained by the public dataset.

point estimate, from which it can sample and characterize the uncertainty of weights. DPS-GLD (Wang et al., 2015) is built to train DP Bayesian networks by adding per-sample clipping. Recent work (Bu et al., 2021) proves that DPSGLD is equivalent to DPSGD with regularization (Bu et al., 2021).

- **DPDFA**. DFA (Nø kland, 2016) emerges as an effective alternative to backpropagation in non-private training. DPDFA (Ohana et al., 2021; Lee & Kifer, 2020) enforces DP in DFA by clipping activations and error signals after the feed-forward phase, scaling the error transport matrix, and adding Gaussian noises to the update direction.

- **PATE**. PATE (Papernot et al., 2018) is a framework based on private knowledge aggregation of an ensemble model and knowledge transfer. It trains an ensemble of *teachers* on disjoint subsets of the private dataset. The ensemble's knowledge is then transferred to a *student* model via differentially private aggregation of the teachers' votes on samples from an unlabeled public dataset. The student model is released as the output of the training.

## 2.2 HANDCRAFTING AND LEARNING LOW-LEVEL FEATURES

**Handcrafted features**. Tramer & Boneh (2021) advocates adopting the wavelet-transform-based ScatterNet (Oyallon & Mallat, 2015; Oyallon et al., 2018) to extract non-learned handcrafted features (e.g., invariance to small rotations and translations) for differentially private learning. A private linear classifier learned on these handcrafted features exhibits higher utility than an end-to-end CNN under a moderate privacy budget.

**Learned features**. In the non-private setting, YM. et al. (2020) demonstrates the features learned from one single image with strong augmentations and via contrastive learning (Chen et al., 2020) can obtain a non-trivial accuracy on ImageNet. However, the performance of these learned features is never studied in the DP domain. In this paper, we show that the learned features are more effective than the handcrafted ScatterNet features in the DP domain.

**Combining handcrafted and learned features**. HarmonicNet (Ulicny et al., 2019) is a data-efficient network using a set of non-learned handcrafted filters based on Discrete Cosine Transform (DCT) at multiple frequencies combined by learnable weights. HarmonicNet obtains a higher non-private accuracy than ScatterNet particularly with small training datasets. However, the performance of the features generated by HarmonicNet is never examined in the DP domain.

## 3 SELF-SUPERVISED PRETRAINING FOR DIFFERENTIALLY PRIVATE LEARNING

In this section, we explain how to build scalable pretrained features for differentially private learning with various sizes of public datasets, i.e., no public dataset, a moderate size (38K∼50K) public dataset, and a large size (>1M) public dataset. Although batch normalization (BN) can be supported by DP in principle, it is difficult to use the same hyper-parameters, e.g., noise multiplier and learning rate, to train BN parameters and the other network parameters in the DP domain. The SOTA DP-enabled training libraries such as Opacus (Yousefpour et al., 2021) do not even support BN yet. Therefore, in this paper, we enforce that DP-enabled training directly happening on the private dataset requires the model to use only group normalization, and all pretrained features are generated with the BN parameters trained by only the public dataset. How to fine-tune BN parameters with private datasets is beyond the topic of this paper, and can be answered by Luo et al. (2021). Results of BN on private datasets can be viewed in Appendix A.2. The utility and privacy improvement of the features generated by SSP over non-learned handcrafted features and those trained with labels cannot be reduced by private BN.

### 3.1 NO PUBLIC DATASET IS AVAILABLE

When no public dataset is available, we compare two ways to train a private model on a private dataset. One way is to train the private model directly on the private dataset using non-learned handcrafted features. The other way is to train a feature extractor by a data-efficient network, self-supervision, and one image, and then to train a private linear classifier on the private dataset using features produced by the feature extractor. We used only DPSGD for DP training in this section.

Table 2: The utility comparison of private models directly trained on a private CIFAR10 under $(\epsilon = 3, \delta = 10^{-5})$.

| Public Dataset | Network Architecture | Acc (%) |
|---|---|---|
| none | end-to-end CNN | 59.2 |
| | ScatterNet+linear | 64.2 |
| | **ScatterNet+CNN** | **66.1** |
| | HarmonicNet + linear | 47.7 |
| | HarmonicNet + CNN | 50.2 |

Table 3: The utility comparison of private linear classifiers learned on pretrained features and a private CIFAR10 under $(\epsilon = 3, \delta = 10^{-5})$.

| Public Dataset | Feature Extractor | | Acc (%) |
|---|---|---|---|
| one image | ResNet18 | ResBlock-1 | 49.7 |
| | | ResBlock-2 | 56.3 |
| | | ResBlock-3 | 51.4 |
| | | ResBlock-4 | 50.3 |
| | | ScatRN18 | 63.1 |
| | | **HarmRN18** | **66.3** |

**Training directly on the private dataset**. We show the utility comparison of various schemes directly learning private CIFAR10 under the privacy budget of $(\epsilon = 3, \delta = 10^{-5})$ in Table 2. The training schemes include a 5-layer CNN, a linear classifier learned on the ScatterNet features, and a 5-layer CNN learned on the ScatterNet features, which are adopted from Tramer & Boneh (2021). We also compare a linear classifier and a 5-layer CNN learned on the DCT-based HarmonicNet handcrafted features. The CNN with the ScatterNet features achieves the highest utility, i.e., 66.1%. We find the HarmonicNet features are vulnerable to noises, i.e., applying noises to the weights combining multiple DCT frequencies significantly degrades the utility of private models.

**Training a feature extractor**. We applied a series of aggressive data augmentations (e.g., cropping) on a single $600 \times 225$ image (shown in Appendix A.3) to create a synthetic dataset as the public dataset. More details of our experimental methodology can be found in Section 4. To first train a feature extractor on the synthetic dataset, we used three different network architectures including ResNet18 (He et al., 2016), ScatRN18, and HarmRN18 to serve as the backbone of the self-supervised SimCLRv2 (Chen et al., 2020). ScatRN18 means ResNet18 with ScatterNet handcrafted features, while HarmRN18 indicates ResNet18 where all convolutional filters are replaced by DCT-based HarmonicNet filters. ResNet18 contains four basic blocks connected sequentially. We extracted pretrained features from all four basic blocks, and use ResBlock-$X$ to represent the pretrained features extracted from the block $X$ of ResNet18. We show the CIFAR10 utility comparison of private classifiers learned on the pretrained features produced by different feature extractors under the privacy budget of $(\epsilon = 3, \delta = 10^{-5})$ in Table 3. All private classifiers have 1 layer. Although compared to the last ResBlock, the two middle basic blocks of ResNet18 produce better features, a single image is not enough to make ResNet18 generate high-quality features. Both ScatRN18 and HarmRN18 output better features than ResNet18. The features produced by HarmRN18 (66.3%) slightly outperform the ScatterNet features (ScatterNet+CNN 66.1% in Table 2) used directly in the DP training, since DCT is fused with every filter of HarmRN18. We will show the HarmRN18 features can obtain a significantly higher utility than the ScatterNet features with the same privacy loss on other complex datasets in Section 5.1.

## 3.2 A MODERATE SIZE PUBLIC DATASET IS AVAILABLE

**Fine-tuning by public unlabeled images**. To further improve the quality of pretrained features, we randomly selected 1K unlabeled image from a public CIFAR100 dataset to fine-tune the feature extractors, as shown in Table 4, where we still used DPSGD to train the private classifier. Training with 1K unlabeled images improves the utility of the private classifier learned on the features generated by HarmRN18 to 72.4%. The entire unlabeled CIFAR100 dataset (50K images) makes the private classifier achieve 78% accuracy on the private CIFAR10 dataset.

**Comparing DP-enabled training frameworks**. We used different DP-enabled training frameworks, i.e., PATE, DPSGD, DPSGLD, and DPDFA, to train a private classifier learned on the HarmRN18 features for private CIFAR10. The HarmRN18 feature extractor is trained by a public CIFAR100 dataset. For PATE, teacher and student models are 1-layer linear classifiers, and use the HarmRN18 features. We adopted 1K teacher models, and CIFAR100 as the public dataset. The private classifiers of DPSGD and DPSGLD have one layer, while the private classifier of DPDFA is a 2-layer multilayer perceptron (MLP), where the first layer uses Tanh activations and the sec-

Table 4: The CIFAR10 utility comparison of the features produced by feature extractors further fine-tuned by different numbers of public images from CIFAR100 under ($\epsilon = 3, \delta = 10^{-5}$).

| Pretrained Feature Extractor | Acc (%) ($\epsilon = 3, \delta = 10^{-5}$) | | | |
|---|---|---|---|---|
| | Unlabeled Image # | | | |
| | 0 | 1K | 10K | 50K |
| ResNet18 | 56.3 | 62.6 | 66.8 | 71.5 |
| ScatRN18 | 63.1 | 67.2 | 70.8 | 73.2 |
| **HarmRN18** | 66.3 | 72.4 | 73.1 | **78** |

Table 5: The CIFAR10 utility comparison of private classifiers learned on the HarmRN18 features and trained by various DP-training methods (DP-SOTA-2: Luo et al. (2021)).

| Public Dataset | Training Scheme | $\epsilon$-DP | Acc (%) |
|---|---|---|---|
| unlabeled CIFAR100 | PATE | 16 | 70.3 |
| | DPSGD | 1.5 | 75.5 |
| | DPSGLD | 1.5 | 74.9 |
| | **DPDFA** | **1.5** | **78.2** |
| CIFAR100 | DP-SOTA-2 | 1.5 | 71 |

ond layer uses a Sigmoid activation. We actually tried a 2-layer MLP for all frameworks, but it works better only with DPDFA, as shown in Appendix A.4. The comparison between DP-enabled training frameworks is shown in Table 5. Compared to DPSGD, the semi-supervised PATE cannot fully take advantage of the pretrained features. The student model has to perform many queries on teacher models to obtain a reasonably high utility, so the privacy loss is high. DPSGLD is similar to DPSGD, but it costs higher privacy overhead in each epoch. As a result, DPSGLD achieves a slightly worse utility than DPSGD under the same privacy budget. Although we used the worst privacy overhead per epoch to estimate the privacy loss of DPDFA, DPDFA still obtains the best utility when $\epsilon = 1.5$ among all frameworks. DPDFA makes the private classifier learned on the HarmRN18 features obtain a much better utility than the classifier (Luo et al., 2021) learned on the features trained by labeled CIFAR100. The comparisons between these DP-enabled training frameworks on more complex private datasets can be found in Section 5.4.

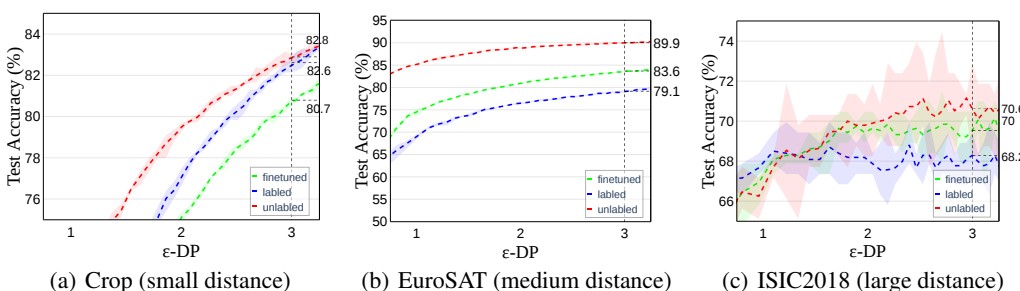

(a) Crop (small distance)     (b) EuroSAT (medium distance)     (c) ISIC2018 (large distance)

Figure 1: The comparison of features trained by labeled and unlabeled mini-ImageNet on various out-of-domain complex private datasets.

**Pretrained features for cross-domain private datasets**. To exam the effectiveness of pretrained features on a private dataset from a different domain, we adopted mini-ImageNet (Vinyals et al., 2016) as our public dataset to train the pretrained features, and studied the utility-privacy trade-off on out-of-domain private datasets such as CropDiseases (Crop) (Mohanty et al., 2016a), EuroSAT (Helber et al., 2019), and ISIC2018 (Tschandl et al., 2018). We first trained HarmRN18 by SimCLRv2 on the mini-ImageNet dataset with no label. And then, we used HarmRN18 as the feature extractor to produce features on private Crop, EuroSAT, and ISIC2018 datasets. Finally, we trained a private classifier on each set of features. We trained a ResNet18 feature extractor (Luo et al., 2021) by the labeled mini-ImageNet dataset as our baseline. Our baseline is trained using the standard cross-entropy loss with label smoothing. We set the label-smoothing parameter to 0.1. The comparison between our scheme and baseline is shown in Figure 1. Except ISIC2018 under $\epsilon < 1.3$, the features generated by SSP enable the private classifier to achieve a better utility than our baseline under the same privacy budget. Particularly, the features produced by SSP on a moderate size of public dataset are already capable enough even for private datasets (e.g., ISIC2018) having a larger learning distance from the public dataset. Compared to supervised pretraining, SSP generates better low- and mid-level features (Zhao et al., 2021), which are more critical to the utility in the DP domain. We also find further fine-tuning these pretrained features with all labels of the public dataset actually degrades the feature quality on these private datasets. The intra-class invariance (Zhao et al., 2021)

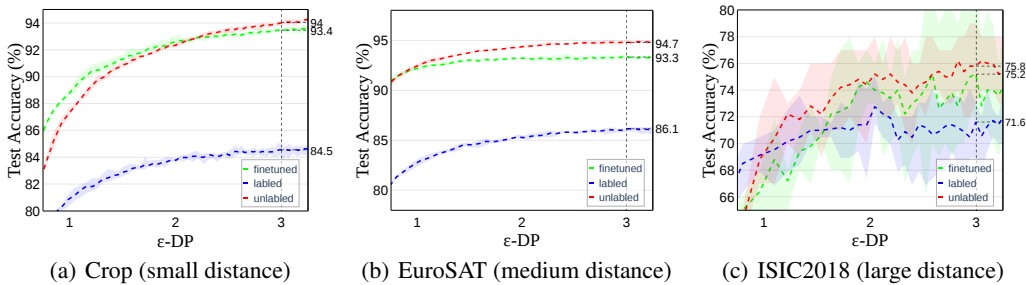

|  (a) Crop (small distance) | (b) EuroSAT (medium distance) | (c) ISIC2018 (large distance) |

Figure 2: The comparison of features trained by labeled and unlabeled ImageNet on various out-of-domain complex private datasets.

introduced by the fine-tuning with all labels of the public dataset increases the class misalignment of private datasets.

### 3.3 A LARGE PUBLIC DATASET IS AVAILABLE

It is easier for a private classifier learned on the features trained by a large public dataset to obtain high utility and small privacy overhead. We adopted ImageNet-1K (Deng et al., 2009) as our public dataset to train features with and without labels. And then, we trained a private classifier using these features on various private complex datasets Crop, EuroSAT, or ISIC2018. We used ResNet50 as the backbone of SimCLRv2, and then the feature extractor for private datasets. We also trained another ResNet50 with labels as our baseline. The comparison between our scheme and baseline is shown in Figure 2. For all three datasets, the features produced by SSP make the private classifier obtain a much higher utility than our supervised-learning baseline under the same privacy budget. A larger unlabeled public dataset greatly improves the quality of features yielded by SSP.

## 4 METHODS

**Public datasets**. We selected the "Image-B" from YM. et al. (2020), which has the size equivalent to $\sim 132$ CIFAR images, due to its rich textures and diversity. The image can be viewed in Appendix A.3. We applied a series of aggressive data augmentations including cropping, scaling, rotation, contrast changes, and adding noise on the image, and then created a synthetic dataset having 50K CIFAR/mini-ImageNet-size images. The detailed parameters of these augmentations can be viewed in YM. et al. (2020). Besides "Image-B", we also used CIFAR100, mini-ImageNet (Vinyals et al., 2016), ImageNet (Deng et al., 2009), and PASS (Asano et al., 2021) as public datasets.

**Private datasets**. We studied multiple private datasets such as CIFAR10/100 (Krizhevsky et al., 2009), CropDiseases (Mohanty et al., 2016b), EuroSAT (Helber et al., 2019), ISIC2018 (Codella et al., 2018), and ImageNet. CropDiseases/EuroSAT/ISIC2018 has a small/medium/large learning distance from the public dataset mini-ImageNet.

**Feature pretraining**. We adopted SimCLRv2 (Chen et al., 2020) to train a feature extractor on different public datasets with no label. The backbone of SimCLRv2 uses a ResNet18, ResNet50, or HarmRN18 architecture. The pretraining of each model lasts 200 epochs with a batch size of 64 or 128. After SSP, we used the backbone as an feature extractor.

**Supervised learning pretraining**. We trained a supervised learning feature extractor on various public datasets using the standard cross-entropy loss with label smoothing. We set the label-smoothing parameter to 0.1.

**DP-enabled training**. We adopted multiple frameworks including DPSGD (Abadi et al., 2016), DPDFA (Ohana et al., 2021; Lee & Kifer, 2020), DPSGLD (Bu et al., 2021) and PATE (Papernot et al., 2018) to train private classifiers. Only DPDFA trains a 2-layer private MLP classifier, while the other DP-enabled training frameworks train a 1-layer linear classifier.

**Normalization layers**. Models learned on handcrafted features and directly trained on private datasets use only group normalization. All pretrained features are produced with the BN parameters trained by only the public dataset. Results related to BN can be found in Appendix A.2.

Table 6: The utility comparison of features trained by a single image.

| Public Dataset | Network Architecture | CIFAR10 | CIFAR100 | Crop | EuroSAT | ISIC2018 |
|---|---|---|---|---|---|---|
| | | | Utility (%) under $\epsilon$-DP= 1 $\vert$ 2 | | | |
| None | ScatterNet + CNN | 51.6 $\vert$ 63.4 | 14.7 $\vert$ 25.7 | 45.1 $\vert$ 67.6 | 17.6 $\vert$ 34.8 | 61.2 $\vert$ 63.7 |
| 1 image | HarmRN18 | **61.1** $\vert$ **64.8** | **22.3** $\vert$ **27.1** | **70.1** $\vert$ **74.3** | **80.1** $\vert$ **84.5** | **65.1** $\vert$ **67.2** |

Table 7: The utility comparison of features trained by a public mini-ImageNet dataset.

| Public Dataset | Extractor Architecture | CIFAR10 | CIFAR100 | Crop | EuroSAT | ISIC2018 |
|---|---|---|---|---|---|---|
| | | | Utility (%) under $\epsilon$-DP= 1 $\vert$ 2 | | | |
| labeled | ResNet18 | 68.8 $\vert$ 72.7 | 33.8 $\vert$ 39.1 | 62.2 $\vert$ 77.0 | 69.3 $\vert$ 76.4 | **68.4** $\vert$ 67.8 |
| unlabeled | ResNet18 | 69.1 $\vert$ 73.0 | **34.9** $\vert$ **40.8** | 75.2 $\vert$ 78.1 | 84.9 $\vert$ 87.6 | 68.1 $\vert$ **70.2** |
| unlabeled | HarmRN18 | **69.5** $\vert$ **73.5** | 34.3 $\vert$ 40.4 | **76.0** $\vert$ **79.5** | **85.2** $\vert$ **88.8** | 67.7 $\vert$ 69.8 |

Table 8: The utility comparison of features trained by a public ImageNet dataset.

| Public Dataset | Extractor Architecture | CIFAR10 | CIFAR100 | Crop | EuroSAT | ISIC2018 |
|---|---|---|---|---|---|---|
| | | | Utility (%) under $\epsilon$-DP= 1 $\vert$ 2 | | | |
| labeled | ResNet50 | 90.4 $\vert$ 91.1 | 61.3 $\vert$ 65.4 | 81.1 $\vert$ 83.7 | 82.9 $\vert$ 85.2 | 69.5 $\vert$ 72.7 |
| unlabeled | ResNet50 | **91.6** $\vert$ **92.7** | **63.3** $\vert$ **69.2** | **87.7** $\vert$ **92.3** | **92.5** $\vert$ **94.3** | **70.3** $\vert$ **75.2** |

**Hyper-parameter search**. Similar to Tramer & Boneh (2021); Luo et al. (2021), we do not count the privacy leakage during hyper-parameter searches on network architectures, optimizers, and hyper-parameters. We target a moderate DP budget of ($\epsilon \leq 4.5, \delta = 10^{-5}$) for private ImageNet, and a small DP budget of ($\epsilon = 0 \sim 2, \delta = 10^{-5}$) for the other private datasets. We fixed the gradient clipping threshold to $C = 0.1$ as default, and tried different batch sizes $|B|$ and learning rates $\eta$. The typical batch size we used is 4096, 8192, or 16384. For each set of hyper-parameter, we ran the experiment for five times and report the average values.

**Library**. We implemented all privacy-related experiments by Opacus v1.1.1 Yousefpour et al. (2021), where native Poisson sampling is supported by BatchMemoryManager. Private models compute per sample gradients, and the DP optimizer does gradient clipping and noise addition.

## 5 RESULTS

To measure a classifier's utility for a range of privacy budgets, we compute the test accuracy and the DP budget $\epsilon$ after each training epoch. For a small DP budget ($\epsilon \leq 2, \delta = 10^{-5}$), we studied the features trained with a single image, the public mini-ImageNet dataset, and the public ImageNet dataset. We also compared various DP-enabled training frameworks to train a private classifier with these features on various private datasets.

### 5.1 NO PUBLIC DATASET

When there is no public dataset, we trained a HarmRN18 network by SimCLRv2 on a single image as our feature extractor. And then, we trained a private 1-layer classifier by DPSGD on various private datasets using features produced by the HarmRN18 feature extractor. As Table 6 shows, the utility of the private classifier is much higher than that of a 5-layer CNN learning directly on the non-learned ScatterNet handcrafted features, when $\epsilon = 1$ and 2. It is difficult to directly learn on even the ScatterNet handcrafted features due to clipped and noisy gradients during DPSGD. The advantage of the features trained by a single image is particularly significant on private datasets like Crop and EuroSAT. This suggests that leveraging features generated by SSP is a scalable solution to differentially private learning even when facing the lack of public datasets.

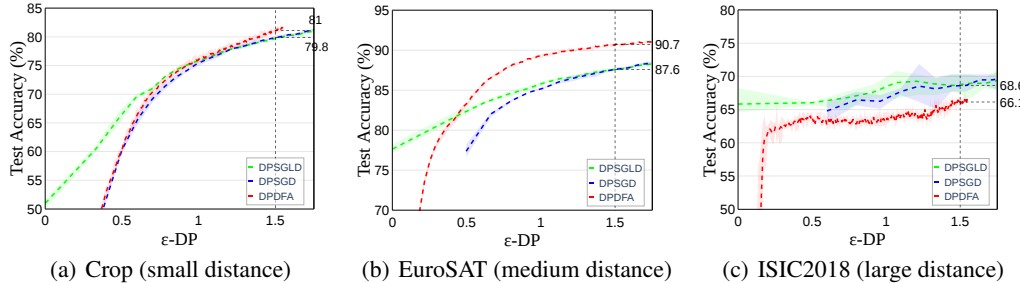

Figure 3: The comparison of DPSGD, DPSGLD, and DPDFA when training with the features produced by a public mini-ImageNet dataset.

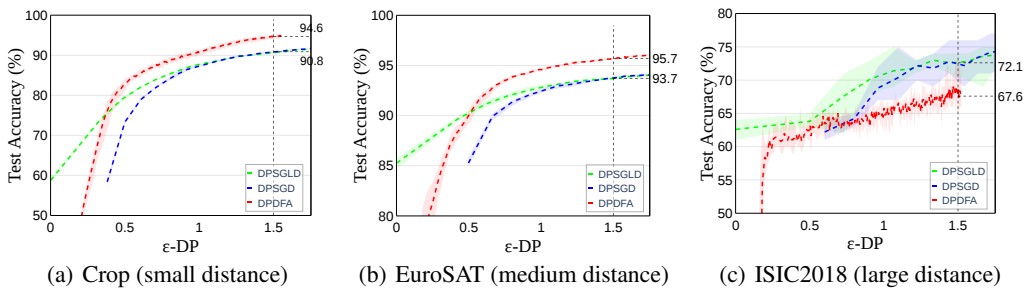

Figure 4: The comparison of DPSGD, DPSGLD, and DPDFA when training with the features produced by a public ImageNet dataset.

## 5.2 A MODERATE SIZE PUBLIC DATASET

When a public mini-ImageNet dataset is available, we trained a HarmRN18 network and a ResNet18 network via SimCLRv2 as our feature extractors. We then used their features to train two private 1-layer classifiers by DPSGD on private CIFAR10, CIFAR100, Crop, EuroSAT, and ISIC2018. Compared to our supervised-learning baseline, as Table 7 shows, the utility of the private classifiers learned on features produced by SSP is higher than that learned with labels except the ISIC2018 dataset having a large learning distance, when $\epsilon = 1$. Although HarmRN18 outperforms ResNet18 in the non-private domain (Ulicny et al., 2019), we do not find there is a significant difference between them in the DP domain. Particularly, the self-supervised pretrained ResNet18 extractor generates slightly better features than HarmRN18 for private CIFAR100 ($\epsilon = 1$ and 2) and ISIC2018 ($\epsilon = 2$) datasets. When pretrained on public mini-ImageNet, ResNet18 is strong enough for producing features for various private datasets.

## 5.3 A LARGE SIZE PUBLIC DATASET

A public ImageNet dataset greatly improves the quality of pretrained features for differentially private learning. We used ResNet50 as the backbone of SimCLRv2 and our feature extractor. And then, we trained a private 1-layer classifier by DPSGD on the features of various private datasets. As Table 8 highlights, the utility of the private classifier learned on features produced by SSP is much higher than that learned with labels, when $\epsilon = 1$ and 2. A large unlabeled public dataset is the key to improving the utility and privacy loss of differentially private learning.

## 5.4 COMPARING VARIOUS DP-ENABLED TRAINING FRAMEWORKS

We compared DPSGD, DPSGLD, and DPDFA to train a private classifier using the ResNet50 features pretrained on public mini-ImageNet and ImageNet for three private datasets including Crop, EuroSAT, and ISIC2018. Since the utility of PATE is much lower than the other three training frameworks under the same DP budget, we excluded PATE in the comparison. DPSGD is the most widely

used DP-enabled training algorithm, so we use it as our baseline. As Figure 3 and 4 highlight, compared to DPSGD, DPSGLD obtains the same utility, but uses only a smaller privacy budget $\epsilon \leq 1$. However, for a large privacy loss $\epsilon > 1$, DPSGD achieves a very similar utility to DPSGLD. DPDFA typically can obtain higher utility than the other two DP-enabled training frameworks on the private datasets having a smaller learning distance from the public dataset, i.e., Crop and EuroSAT, under the same privacy budget when $\epsilon > 0.5$. However, for private datasets having a large learning distance from the public dataset, i.e., ISIC2018, DPDFA achieves only a lower utility than DPSGD and DPSGLD under the same privacy budget.

## 6 CONCLUSION AND FUTURE WORK

**Conclusion**. We have demonstrated that SSP is a simple yet scalable solution to differentially private learning regardless of the size of available public datasets. The features produced by SSP on a single image, or a moderate/large size public dataset significantly outperform the features trained with labels in the DP domain, let alone the non-learned handcrafted ScatterNet features. Based on the learning distance from the public dataset and the privacy budget, different private datasets may favor distinctive DP-enabled training frameworks to train their private classifiers learned on features produced by SSP.

**Future work**. We also tried SSP on private ImageNet-1K and obtained a top-1 accuracy of 25.3% when $\epsilon = 3$. However, more research efforts are required to further improve the utility of large-scale private datasets under a moderate privacy budget ($\epsilon \leq 3$). We adopted ResNet50 as the backbone of SimCLRv2, and trained it on a public PASS dataset consisting of 1.4 million CC-BY images with no label. The backbone is used to generate features for private ImageNet. The features can yield 60.8% top-1 non-private accuracy. We further trained a private classifier on these features in the DP domain. We find that the gradient clipping threshold $C = 0.1$ only produces a top-1 utility 8.1% even when $\epsilon = 3$. So we enlarged $C$ to 1, and adjusted the other

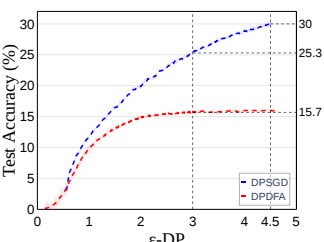

Figure 5: The utility and privacy loss of private ImageNet-1K.

DP parameters. The utility and privacy overhead of the classifier on private ImageNet is shown in Figure 5. When $\epsilon = 3$, the top-1 utility of the classifier becomes 25.3%. We also compared DPSGD and DPDFA to train our private classifier. Since PASS is still too small to produce strong enough pretained features, DPDFA achieves a lower utility than DPSGD under the same privacy budget. Although Kurakin et al. (2022) uses supervised-learning-based features to achieve a top-1 utility 47.9% on private ImageNet with $(\epsilon = 10, \delta = 10^{-6})$, its top-1 utility is only 7.6% when $\epsilon = 4.57$. Under a moderate privacy budget ($\epsilon \leq 4.5$), the features generated by SSP greatly outperform prior supervised-learning-based features. More recently, Mehta et al. (2022) obtains a top-1 utility of 81.5% on private ImageNet under $\epsilon = 2$ by features trained by JFT-300M and JFT-4B datasets (Sun et al., 2017). However, JFT-300M and JFT-4B datasets are not publicly available. We leave further improving the utility of private ImageNet under a small or moderate privacy budget to future work.

## 7 ETHICS STATEMENT

Our study improves the privacy of sensitive training data in various machine learning models. If our proposed technique fails, the privacy of sensitive training data will not be worse than our baseline.

## 8 REPRODUCIBILITY STATEMENT

Our code is anonymously released at `https://anonymous.4open.science/r/noname/`.

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

# A APPENDIX

## A.1 RÉNYI DP

Rényi DP (RDP) (Mironov, 2017) is a generalization of $(\epsilon, \delta)$-DP that uses Rényi divergence as a distance metric. The Rényi divergence of order $\alpha$ between two distributions $P$ and $Q$ is defined as:

$$D_\alpha(P||Q) = \frac{1}{\alpha - 1} \log \mathbb{E}_{x \sim P} \left[ \left( \frac{P(x)}{Q(x)} \right)^{\alpha-1} \right]$$

A model satisfies $(\alpha, \epsilon)$-RDP if

$$D_\alpha(M(D)||M(D')) = \frac{1}{\alpha - 1} \log \mathbb{E}_{x \sim M(D)} \left[ \left( \frac{\mathbf{Pr}[M(D) = x]}{\mathbf{Pr}[M(D') = x]} \right)^{\alpha-1} \right]$$

Pure $(\epsilon, 0)$-DP is equivalent to $(\infty, \epsilon)$-RDP. And if a model $M$ satisfies $(\alpha, \epsilon)$-RDP, $M$ also satisfies for any $\delta \in (0, 1)$. RDP during the training of a model is enforced by two components: per-sample gradients are clipped at a fixed L2 norm threshold $C$, and Gaussian noise of magnitude $\sigma^2 C^2$ is added to the gradient updates for a noise scale parameter $\sigma$.

Table 9: The utility and privacy comparison of various schemes using batch normalization on a private CIFAR10 dataset. DP-SOTA-1: Tramer & Boneh (2021); and DP-SOTA-2: Luo et al. (2021).

| Public Dataset | Scheme | $\epsilon$-DP | Network | Training | Accuracy (%) |
|---|---|---|---|---|---|
| None | DP-SOTA-1 | 3 | Scat + CNN | DPSGD | 69.3 |
| 1 image | Ours | 3 | HarmRN18 | DPSGD | 71.1 |
| labeled CIFAR100 | DP-SOTA-2 | 1.5 | ResNet18 | DPSGD | 81.6 |
| unlabeled CIFAR100 | Ours | 1.5 | ResNet18 | DPSGD | 80.7 |

## A.2 PRIVATE DATA NORMALIZATION

Prior work (Tramer & Boneh, 2021; Luo et al., 2021) finds batch normalization (BN) greatly improves convergence on complex private datasets. Private BN (Tramer & Boneh, 2021) is proposed to compute private estimates of the per-channel mean and variance of the ScatterNet features. It is difficult to use the same set of hyper-parameters to train BN layers and the other private components in a neural network. Different values for noise multi-plier (Tramer & Boneh, 2021) or learning rate (Luo et al., 2021) are used to train BN layers. More-over, most DP training libraries do not support BN layers in a private model yet. We adopt the same method and hyper-parameters to train BN layers as prior work (Tramer & Boneh, 2021; Luo et al., 2021). The utility and privacy comparison between our schemes and prior work is shown in Table 9. The features trained by HarmRN18 using BN on a single image obtain better utility than the ScatterNet non-learnable handcrafted features. However, the features trained by ResNet18 on a public unlabeled CIFAR100 achieve a slightly worse utility than those trained with labels (Luo et al., 2021). This is because besides BN layers, (Luo et al., 2021) also fine-tunes convolutional layers in the DP domain.

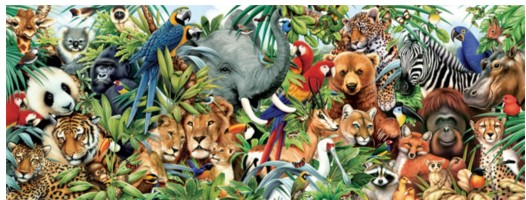

Figure 6: Single-image self-supervision.

## A.3 LEARNING ON A SINGLE IMAGE

Recent work (YM. et al., 2020) shows the self-supervised learning methods such as BiGAN, RotNet, and DeepCluster can be used to train the first few layers of a deep network model using a single training image, when sufficient data augmentation is used. We select the "Image-B", which is shown as Figure 6, due to its rich texture and high diversity. The image size is $600 \times 225$. We also try the "Image-A" and the "Image-C" from (YM. et al., 2020) in our experiments. Among three images, "Image-B" achieves the best utility and privacy budget for most private classifiers.

Table 10: The utility comparison of classifiers consisting of different numbers of layers.

| Public Dataset | Private Dataset | Arch + Train | $\epsilon$ | Utility (%) |
|---|---|---|---|---|
| CIFAR100 | CIFAR10 | 1-layer DPSGD | 1.5 | 75.5 |
| | | 2-layer DPSGD | 1.5 | 73.3 |
| | | 1-layer DPSGLD | 1.5 | 74.2 |
| | | 2-layer DPSGLD | 1.5 | 72.8 |
| | | 1-layer PATE | 16 | 70.2 |
| | | 2-layer PATE | 16 | 64.7 |
| | | 2-layer **DPDFA** | 1.5 | **78.2** |

## A.4 SINGLE-LAYER AND 2-LAYER CLASSIFIERS

In order to study DPDFA, we need to use a 2-layer linear MLP to serve as our private classifier, where the first layer uses Tanh activations and the second layer uses a Sigmoid activation. A natural question is "how does DPSGD work with a 2-layer MLP?". As Table 10 shows, we find that unlike DPDFA, all the other DP-enabled training frameworks work better with a single-layer linear classifier. For instance, a 2-layer MLP-based private classifier is always overtopped by a 1-layer classifier when trained by DPSGD. Therefore, in this paper, we always adopt a 2-layer classifier for DPDFA, and a 1-layer classifier for the other DP-enabled training frameworks.

