# OpenReview forum: "SELF-SUPERVISED PRETRAINING FOR DIFFERENTIALLY PRIVATE LEARNING"
_ICLR.cc/2023/Conference — Submitted to ICLR 2023_

### Official Review · Reviewer_SheV · 2022-10-23

**Confidence:** 4
**Correctness:** 2
**Technical Novelty And Significance:** 2
**Empirical Novelty And Significance:** 2
**Recommendation:** 3

**Clarity, Quality, Novelty And Reproducibility:**

## Clarity

The paper is easy to read.

## Quality

I have explained that I find the claims to not be well supported.

## Novelty

As is, I do not think the claims are very novel. But that is mostly because there isn't a clear discussion of what are the main claims of the paper and how they are validated.

## Reproducibility

I haven't run the code but I have noticed that a link is provided to an anonymous repository.

**Strength And Weaknesses:**

## Strength

* The paper addresses and important and relevant topic of using public data to boost the accuracy of privately learning. There are a wide range of options for how to use the public data, the paper provides an insight different types of such algorithms

* The paper also experiments with various types of private training algorithms and various types of public and private datasets. This is perhaps one of the more extensive experimental studies I have seen in this domain for computer vision.

## Weakness

* I am missing the main message the paper wants to convey. It appears to me as if it is a repository of experimental findings, which is nice in its own right, but the experimental findings suffer from a lot of confounders and needs a systematic study to draw meaningful conclusions. I have given examples in my summary.
* The paper also doesn't provide reasons/explanations for many of its claims/observations. For example, why scatternet is better than HarmNet in Table 1., why DPDFA is better than others in table 5. While a rigorous theoretical exposition is not always necessary, in my opinion, there needs to be an explanation and a validation of the explanation.

Thus despite having a lot of results, I cannot recommend acceptance of the paper unless the study is made more systematic and proper explanations are provided.

**Summary Of The Paper:**

The paper presents various ways of using Self-Supervised pre-training (SSP) using public data in the context of differentially private learning. In particular, the paper claims to provide suggestions on what is the best approach when varying **amounts** and **types** of public data are available for SSP.

**Summary Of The Review:**

* I think it would be better to use a more relevant citation when referring to Differential Privacy than Abadi et. al. For example Dwork et. al. would be a more relevant citation here.

* It is now quite well known that hyper-parameter selections using the private dataset can lead to privacy leaks. However, usually that is chosen from a relatively small set eg. learning rate is chosen from a few values. However, I wonder whether choosing the public image as Image (b) can lead to a huge leak. I am not sure whether this is indeed true but I would appreciate if the authors provided a discussion of this.

* The conclusion drawn from table 2 is that ScatterNet features are better than those from harmonicNet and the presented reason is that "applying weights to combining multiple DCT frequencies significantly degrades the utility of private models" But this seems rather arbitrary.
    * First, there is no discussion of why we should use HarmonicNet to compare with and why larger models like ResNet and WideResNet are not compared with here.
    * Second, if harmonic net is indeed to be used, why are the features of HarmonicNet more affected than ScatterNet? The presented reason is not backed up with any evidence.
* In Table 5, DPDFA is championed to be the best method. But there are various confounders here. To begin with, DPDFA uses 2 layer MLPs whereas the others use one layer linear as stated in the paper. Second, there are optimisation differences between a method like DPDFA and other methods as the hyper-parameters are very different. I may have missed it but I am not sure how the noise rate and the number of steps of DP-SGD were selected. It is possible that the advantage of DPDFA comes merely from the ease of its hyper-parameter selection. While this is possibly an interesting outcome, it is not discussed in the paper where the advantage of DPDFA comes from.

* In Figure 1 (a) unlabelled SSP and labeled R18 supervised features achieved nearly the same accuracy at $\epsilon=3$ with fine-tuned lacking behind a little bit. On ISIC, there is really no statistically significant difference between the methods. Using these observations the author claims that SSP is better than labeled and in fact, fine-tuned features are worse. This is despite the fact that SSP and supervised features are nearly the same in Figure 2(a) for larger $\epsilon$ and in Figure 2(b) for small $\epsilon$ and figure 2(c) almost everywhere. Moreover, in Figure 2, when using the full imagenet, this trend is reversed. Fine-tuned features do better than SSP and much better than Supervised R18 feature. But the text in section 3.3 claims the opposite that SSP features are better. Thus, there are a lot of experiments here but I can't find a hypothesis that can be validated or negated using them. The claims made in the paper do not hold consistently across the experiments in the paper.

* An interesting question from Figure 2, according to me, is why are the accuracies from supervised R18 and finetuned features so different ?

* From Figure 3,4 the paper claims that DP-DFA is better than the other methods when there is a small distance between pre-training and fine-tuned distance. However, it seems quite anecdotal without a systemic study of varying this distance. Perhaps, the authors can use the setting of _Controllable-Shift (CS) datasets_ from Shi et. al. 2022.


Dwork, Cynthia, et al. "Calibrating noise to sensitivity in private data analysis." Theory of cryptography conference. Springer, Berlin, Heidelberg, 2006.
Shi, Yuge, et al. "How robust are pre-trained models to distribution shift?." arXiv preprint arXiv:2206.08871 (2022)

---

### Official Review · Reviewer_wJNa · 2022-10-24

**Confidence:** 4
**Clarity, Quality, Novelty And Reproducibility:** The paper is clearly written and the …
**Correctness:** 3
**Technical Novelty And Significance:** 2
**Empirical Novelty And Significance:** 3
**Recommendation:** 3

**Details Of Ethics Concerns:**

The paper studies how the privacy is protected through self-supervised pretraining on public data. No more concerns

**Strength And Weaknesses:**

The paper studies an important problem that how self-supervised pretraining helps differentially private machine learning. They evaluate the self-supervised pretraining under various sizes of available public datasets. Especially, given one single image, they conduct self-supervised pretraining, which is interesting.

Weakness

The information that pretraining helps differentially private deep learning is not new. Many recent papers reveal this information, which are not covered in the paper.
De et al. 2022 Unlocking High-Accuracy Differentially Private Image Classification through Scale
Yu et al. 2021 Large Scale Private Learning via Low-rank Reparametrization
Yu et al. 2022 Differentially Private Fine-tuning of Language Models
Li et al. 2022 Large Language Models Can Be Strong Differentially Private Learners

For the one image case, the improvement is minimal and the result is not compared with state-of-the-art performance (De et al. 2022) on CIFAR10. Moreover, for the lack of public data, it seems that one has to choose one image carefully (like the Figure 6 in the paper for good utility). Therefore it is  in general hard to say the self-supervised pretraining really helps for the case of lacking of public data.






**Summary Of The Paper:**

The paper demonstrates that self-supervised pretraining (SSP) can help differentially private deep learning regardless of the size of available public datasets in image classification. Specifically, they show the features generated by SSP on only one single image enable a private classifier to obtain better utility than the handcrafted features.

**Summary Of The Review:**

The paper discussed one important topic, but the information has been conveyed in previous work. Moreover the paper does not show convincing result showing self-supervised pretraining helps in the case of public dataset not available. The paper evaluated various network structures and several optimizers, but it does not convey useful and consistent information that would help the choice in practice. I would not recommend its acceptance.

---

### Official Review · Reviewer_YM6J · 2022-10-25

**Confidence:** 3
**Correctness:** 3
**Technical Novelty And Significance:** 2
**Empirical Novelty And Significance:** 2
**Recommendation:** 5

**Clarity, Quality, Novelty And Reproducibility:**

The clarity can be greatly improved, especially on the structure of the paper. See above for more details.

**Strength And Weaknesses:**

## Strengths:
1. The proposed method is simple yet effective
2. Experiments are comprehensively conducted on a variety of datasets, different privacy constraints, and different DP training schemes.

## Weakness:
1. The novelty is limited. While I acknowledge the authors’ efforts in comparing different DP training methods and conducting experiments on different datasets, the proposed self-supervised learning methods are mostly known methods, which are simple adaptations and applications to the differentially private learning domain.
2. The paper is poorly written and hard to follow. The structure of the paper can be better organized: current Section 4 can be put earlier and better called “experimental setup”; while Section 5 and Section 3 can probably be combined to cover deeper analysis for different data assumptions. The presentation of the tables can be further improved: some tables contain eps=1, 2; while some tables cover eps=3 even given the same dataset. The terms “labeled CIFAR100” and “unlabeled CIFAR100” are also a bit misleading: it is better to distinguish two by “supervised” or “unsupervised” to highlight the method differences.


**Summary Of The Paper:**

This paper focuses on improving the accuracy of differentially private learning accuracy by public self-supervised pre-training. Specifically, the authors propose to perform self-supervised pre-training based on different assumptions regarding the availability of public data: (1) one public image pretraining can outperform handcrafted features under the same privacy budget; (2) when larger public datasets are presented, the features pretrained with self-supervision signals outperform supervised learned features, and demonstrate better performance than existing state-of-the-art DP baselines with different DP training schemes.

**Summary Of The Review:**

Strengths:
1. The proposed method is simple yet effective
2. Experiments are comprehensively conducted on a variety of datasets, different privacy constraints, and different DP training schemes.
Weakness:
1. The novelty is limited.
2. The paper is poorly written and hard to follow.

---

### Official Review · Reviewer_85CV · 2022-10-26

**Confidence:** 4
**Correctness:** 2
**Technical Novelty And Significance:** 2
**Empirical Novelty And Significance:** 2
**Recommendation:** 3

**Clarity, Quality, Novelty And Reproducibility:**

Clarity:
- Each individual section is easy to understand. However overall, paper looks like a collection of loosely related observations. For example, different parts of the paper uses different models, different pre-training datasets, etc…. Results are spread across section 3 and section 5 which makes it hard to read and compare.

Quality:
- I think paper would benefit from better organized experiments and from better comparison with state of the art pre-training methods on labeled data.

Novelty:
- Overall, public pre-training followed by DP finetuning was already studied in prior work. However, none of the prior work tried to use self-supervised learning for pre-training and none of the prior work rigorously studied the relationship between amount of public data and the final result of DP finetuning. Overall I would describe it as a limited amount of novelty.

Reproducibility:
- Authors provide anonymized code and describe experiments with sufficient details. This should be enough to reproduce their results, however I haven’t actually tried to run their code.


**Strength And Weaknesses:**

Strength:
- The questions about size of pre-training dataset and pre-training method are important.
- To the best of my knowledge, none of the prior work tried to use self-supervised learning on public data to learn features for private fine-tuning.


Weaknesses:
- Poorly organized experiments. Experimental results are spread across sections 3 and 5, which makes it hard to read and compare.
- Authors actually missing some of the state of the art results in DP training. In particular, CIFAR10 and CIFAR100 results from https://arxiv.org/pdf/2204.13650.pdf are quite a lot better, compared to what is reported in the paper. If results from https://arxiv.org/pdf/2204.13650.pdf are taken into account, then experiments from the paper are not enough to support the statement “SSP greatly outperform the features trained with labels”.


**Summary Of The Paper:**

The paper studies performance of a differentially private model, which is trained on top of a feature extractor. Paper considers feature extractors which are trained using self-supervised learning techniques on public datasets. Paper studies how the amount of public data for pre-training affect performance of the model on the final task.

**Summary Of The Review:**

Reject.

Overall the paper studies interesting and important topics. However paper would be significantly improved with better comparison with other state of the art methods, as well as better organization of experiments within the paper.

---

### Decision · Program_Chairs · 2023-01-20

**Decision:**

Reject

**Justification For Why Not Higher Score:**

Limited novelty and no author responses to the critiques.

**Justification For Why Not Lower Score:**

N/A

**Metareview: Summary, Strengths And Weaknesses:**

Reviewers agreed that the problem studied is interesting. But they pointed out issues related to limited novelty, and lack of comparison/knowledge of several recent works in the area. The authors did not respond to the reviews, so there was minimal follow-up discussion.